# Untangling Macropore Formation and Current Facilitation in P2X7

**DOI:** 10.3390/ijms241310896

**Published:** 2023-06-30

**Authors:** Federico Cevoli, Benoit Arnould, Francisco Andrés Peralta, Thomas Grutter

**Affiliations:** 1Équipe de Chimie et Neurobiologie Moléculaire, Laboratoire de Conception et Application de Molécules Bioactives (CAMB) UMR 7199, Centre National de la Recherche Scientifique, Faculté de Pharmacie, Université de Strasbourg, 67401 Illkirch, France; cevolifederico@gmail.com (F.C.); arnould.benoit38@gmail.com (B.A.); francisco.andres.peralta@gmail.com (F.A.P.); 2Department of Chemistry, Washington University in St. Louis, St. Louis, MO 63130, USA; 3Instituto de Neurociencias, CSIC-UMH, 03550 San Juan de Alicante, Spain; 4University of Strasbourg Institute for Advanced Studies (USIAS), 67000 Strasbourg, France

**Keywords:** P2X7, current facilitation, ATP sensitization, macropore formation

## Abstract

Macropore formation and current facilitation are intriguing phenomena associated with ATP-gated P2X7 receptors (P2X7). Macropores are large pores formed in the cell membrane that allow the passage of large molecules. The precise mechanisms underlying macropore formation remain poorly understood, but recent evidence suggests two alternative pathways: a direct entry through the P2X7 pore itself, and an indirect pathway triggered by P2X7 activation involving additional proteins, such as TMEM16F channel/scramblase. On the other hand, current facilitation refers to the progressive increase in current amplitude and activation kinetics observed with prolonged or repetitive exposure to ATP. Various mechanisms, including the activation of chloride channels and intrinsic properties of P2X7, have been proposed to explain this phenomenon. In this comprehensive review, we present an in-depth overview of P2X7 current facilitation and macropore formation, highlighting new findings and proposing mechanistic models that may offer fresh insights into these untangled processes.

## 1. Introduction 

P2X7 receptors (P2X7) are part of the purinergic family, which consists of ATP-gated P2X receptors (P2X). In mammals, this family comprises seven subunits, namely P2X1 to P2X7, which assemble in the membrane to form homo- or heterotrimeric channels [1,2,3,4,5]. Among P2X receptors, P2X7 demonstrates a unique low sensitivity to ATP. It requires unusually high millimolar-range concentrations of ATP to become activated. As these elevated concentrations are typically found at sites of cell injury, P2X7 is therefore referred to as a “danger-sensing receptor” [6]. Consequently, P2X7 is widely expressed in immune cells, including macrophages and microglia, which are capable of detecting the abnormal release of extracellular ATP (eATP). 

It has long been shown that eATP can act as a signaling molecule [7]. Despite having a short half-life of a few seconds at therapeutic concentrations (0.1 mM in human airways) [8], it is often associated with relatively long-lasting events, particularly those related to inflammatory or pro-inflammatory cellular states [9]. Consequently, P2X7 is involved in several physiological and pathological processes, including inflammation, immune responses, cell proliferation and programmed cell death, such as apoptosis and pyroptosis. In the nervous system, for example, P2X7 has been shown to play a crucial role in various neuropathological conditions, including neurodegeneration, chronic pain and brain injury [10]. The accumulation of eATP at inflammatory sites triggers a range of pathophysiological responses via P2X7, with the most notable being the activation of the NOD-, LPR- and pyrin domain-containing protein 3 (NLRP3) inflammasome in the cytoplasm of mononuclear and polymorphonuclear phagocytes [6]. In the context of neuroinflammation, P2X7-driven inflammasome activation in microglia initiates a cascade of events leading to the production and release of several pro-inflammatory cytokines, such as interleukin-1 beta (IL-1β) as well as reactive oxygen species (ROS) [11]. Furthermore, P2X7 has been implicated in the pathogenesis of chronic pain states [12], where its activation triggers the development and maintenance of conditions such as neuropathic pain, inflammatory pain and cancer pain [13,14]. Due to its ability to induce the release of pro-inflammatory cytokines that may eventually lead to cell death, P2X7 is considered a promising drug target for numerous pathophysiological conditions, particularly in the field of neuroimmunology [15,16,17,18,19,20,21]. 

When eATP binds to P2X7, it triggers the rapid opening of the transmembrane channel, enabling the passage of small monovalent ions, such as sodium (Na^+^) and potassium (K^+^), as well as divalent cations such as calcium (Ca^2+^), across the cell membrane. Unlike other P2X receptors, P2X7 typically does not undergo channel desensitization, which is a temporary inactivation that terminates ion flux despite ATP remaining bound to the receptor. Instead, P2X7 undergoes a facilitation process where currents progressively increase with repetitive or prolonged agonist application [1,22]. Stimulation of P2X7 also leads to the formation of a phenomenon known as P2X7 macropore formation, which involves the permeabilization of the cell membrane to high molecular weight species [22,23]. While other P2X receptors also exhibit permeability to organic cations [24,25,26], the distinct feature of P2X7 macropore formation lies in its ability to allow the passage of nanometer-scale molecules [27]. Macropore formation has also been implicated in cell death mechanisms and certain pathological states, including chronic pain [12]. Despite several hypotheses proposed to explain macropore formation and current facilitation [23,24,25,28], the underlying mechanisms remain poorly understood. 

This article provides a review of our current knowledge regarding current facilitation and macropore formation, while also discussing new findings that may offer fresh insights into these untangled processes. New mechanistic models are proposed, which could contribute to future research directions. 

## 2. P2X7 Structures 

Although the first atomic resolution structure of P2X receptors was published in 2009 [29], the pioneering structures of P2X7 were not released until 2016. These structures were resolved using X-ray crystallography and were based on the receptor obtained from the giant panda [30]. These structures confirmed the trimeric architecture of the ion channel and exhibited an overall structural similarity to previously crystallized P2X receptors [29,30,31,32], with each subunit resembling the characteristic dolphin-like shape. They also confirmed the presence of three ATP-binding sites at the subunit boundaries in the extracellular domain and the presence of two transmembrane segments (TM1 and TM2 from each subunit) that form the ion-conducting pore. However, these structures were highly truncated in both the N- and C-termini due to the requirement of the crystallization process. Since the intracellularly located C-terminus is significant for P2X7, these initial structures provided limited information about its cytoplasmic domain. Notably, P2X7 possesses a unique and unusually large C-terminus that is not present in other determined or predicted P2X structures [33]. Importantly, this C-terminus plays a crucial role in initiating apoptosis [22,34,35,36,37,38,39]. Consequently, in 2019, Mansoor’s group successfully determined the full-length structure of rat P2X7 [40] using single-particle cryo-electron microscopy (cryo-EM). This groundbreaking accomplishment provided invaluable insights into the structural characteristics of P2X7 and significantly advanced our understanding of this receptor. The full-length structure was solved in both the absence (apo state) and the presence of ATP (ATP-bound state), revealing novel structural elements in the cytoplasmic domain (Figure 1A). In comparison to other P2X receptors [4], the cytoplasmic domain of P2X7 significantly protrudes into the cytosol and features two unexpected components: the “C-cys anchor” and the “ballast” (Figure 1B). The C-cys anchor is a cysteine-rich loop consisting of 18 amino acid residues [40,41]. It serves as a connection between TM2 and the cytoplasmic cap, a highly intertwined element found in P2X3 receptors, which is thought to play a role in channel desensitization [32]. The C-cys anchor contains several palmitoylated residues, and it has been demonstrated that it prevents channel desensitization by physically anchoring the palmitoylated groups to the membrane, thus restricting the movement of the cytoplasmic cap [40]. The second element is the ballast, located beneath the cytoplasmic cap and C-cys anchor, which folds into an autonomous domain without any structural homologue in the Protein Data Bank. Surprisingly, three GDP-binding sites and three dinuclear Zn^2+^-binding sites have been discovered within the ballast. However, the role of the ballast, GDP and Zn^2+^ remains completely unknown. Notably, the presence of an intracellularly located GDP-binding site gives the P2X7 the appearance of having two nucleotide-binding sites emerging from the membrane—one on the extracellular side and the other on the intracellular side (Figure 1A).

## 3. Current Facilitation

P2X7 current facilitation occurs when ATP applications are repeated or prolonged (Figure 2A). It manifests as a slow onset of ATP-induced currents, resulting in an increase in both current amplitude and activation kinetics [22,42,43]. At the early stage of ATP application, the inward currents recorded through patch-clamp electrophysiology are typically small compared to those observed at a later stage (around 20 s), where larger inward currents are recorded. Current facilitation generally leads to a leftward shift in the agonist concentration–response curve, indicating higher ATP sensitivity [42,43]. Several factors modulate current facilitation, including cholesterol [44,45,46,47], palmitoylation of residues in the C-cys anchor [46], cooperative interactions between the juxta-transmembrane N- and C-termini [48], Ca^2+^-independent phospholipase A2 PLA2 and chloride (Cl^−^) channels [49]. It is worth noting that modulation by Ca^2+^-dependent calmodulin was also observed for P2X7 [42] but appears to be specific to rat P2X7 and not human P2X7 [50]. While the C-cys anchor plays a crucial role in current facilitation, the removal of the ballast did not affect facilitation currents [40], suggesting that the ballast has a lesser impact on the process of current facilitation.

Several mechanisms have been proposed to explain current facilitation in P2X7. Initially, the pore dilation theory was postulated, suggesting that current facilitation resulted from the dilation of the ion channel pore, which was also proposed to explain macropore formation [24,25]. However, upon closer examination, it has been found that the interpretation of the data supporting the pore dilation theory was deemed unsatisfactory (see next chapter) [54]. Subsequently, new mechanistic insights have emerged. Here, we summarized these findings into two comprehensive models, as depicted in Figure 2B,C.

### 3.1. Model #1: Current Facilitation Requires the Involvement of an Additional Channel

Model #1 suggests that current facilitation occurs through the secondary activation of Cl^−^ channels (Figure 2B). In heterologous systems such as *Xenopus* oocytes, the ectopic co-expression of P2X7 and TMEM16A (or anoactamin-1), a calcium-activated chloride channel (CaCC), leads to a significant and sustained increase in Cl^−^ conductance in response to the Ca^2+^ influx induced by P2X7 opening [51]. It has been suggested that the increasing inward currents, recorded by two-electrode voltage-clamp electrophysiology, represent the combination of two major currents: the P2X7-mediated influx of Na^+^ and Ca^2+^ and the CaCC-mediated efflux of Cl^−^ [51]. Both currents contribute to the downward deflection of the current trace, as the efflux of negative charges is electrically equivalent to the influx of positive charges. Similar results have been obtained in Axolotl oocytes, which lack endogenous CaCCs, thus ruling out the contribution of endogenous channels [51]. However, the impact of the secondary activation of Cl^−^ channels on the modulation in ATP sensitivity in P2X7 remains unknown. Furthermore, in other cell types such as human macrophages, the role of Ca^2+^ as a second messenger in the secondary activation of TMEM16A seems to be less significant. Instead, it has been demonstrated that current facilitation requires the involvement of Cl^−^ channels through a phospholipase A2 (PLA2)-dependent pathway [49]. PLA2 is an enzyme that catalyzes the release of membrane phospholipids upon activation, and it operates independently of Ca^2+^. Certain phospholipids, including 1-palmitoyl-2-oleoyl-sn-glycero-3-phospho-(1′-rac-glycerol) (POPG) and sphingomyelin, have been shown to enhance P2X7 activity [46]. The released phospholipids from PLA2 activation can provide feedback to P2X7, resulting in current facilitation [49]. It is also possible that a similar mechanism affects the unidentified Cl^−^ channel in macrophages. However, the relationship between PLA2 and the Cl^−^ channel is unclear and requires further investigation. 

### 3.2. Model #2: Current Facilitation Is an Intrinsic Property of P2X7

Another recently proposed model, referred to as model #2, suggests that current facilitation is an intrinsic property of the P2X7 itself [47,53] (Figure 2C). Single-channel recordings of rat P2X7 currents in transfected HEK-293 cells have revealed that current facilitation increases the probability of the channel being open [47]. This increase was also observed when cholesterol was acutely removed from the cell membrane that had not been previously exposed to ATP, implying that the binding of cholesterol to P2X7 suppresses channel activity [47]. There is evidence suggesting that cholesterol inhibits P2X7 activity through direct interactions with the transmembrane domain, and the palmitoylated groups of the C-cys anchor residues counteract this inhibitory effect [46]. Deletion of the C-cys anchor or mutation of the palmitoylated residues to alanine both lead to channels that almost completely desensitize, resulting in the loss of current facilitation [40]. This suggests that the C-cys anchor plays a critical role in determining these functional properties [40]. Additionally, a more recent study utilizing voltage-clamp fluorometry, which incorporated the fluorescent unnatural amino acid residue l-3-(6-acetylnaphthalen-2-ylamino)-2-aminopropanoic acid (ANAP), provided compelling evidence that current facilitation is an intrinsic property of P2X7 [53]. By specifically incorporating ANAP into various sites of P2X7, including position F11 in the cap domain (depicted in Figure 1B), the authors identified distinct fluorescent changes that mirrored current facilitation (as shown in Figure 2C). Taken together, these findings suggest that current facilitation is an inherent characteristic of the P2X7 ion channel. 

It is important to acknowledge that two mechanistic models, model #1 and model #2 described here, are not mutually exclusive. It is reasonable to assume that both mechanisms can coexist within cells, albeit with varying degrees of contribution. The relative significance of these mechanisms is likely influenced by regulatory factors that differ from cell to cell. These factors include the composition of the plasma membrane, the levels of endogenous CaCCs and PLA2 and the metabolic state of the cell in relation to cholesterol. Consequently, these regulatory factors may favor one or both mechanisms, further complicating an understanding of current facilitation. A potential future direction of research would be to investigate and evaluate the relative impact of these regulatory factors on current facilitation.

## 4. Macropore Formation

First described in mast cells in 1979 [55], the formation of macropore was initially attributed to the “P2Z” receptor before being reclassified as P2X7 after its cloning [22]. This phenomenon remains one of the most enigmatic and challenging features of P2X7 [23]. While it has been extensively observed in various cell types, both native and heterologous systems, the molecular mechanism underlying macropore formation is still not well understood [41,49,56,57,58,59]. Often confused with current facilitation, several hypotheses have been proposed to explain macropore formation, but none have been entirely satisfactory [23,28]. 

Two primary methodologies have been employed to study macropore formation. The first involves monitoring the cellular uptake (or efflux from preloaded cells) of fluorescent dyes upon P2X7 activation (Figure 3A). The most commonly used fluorescent dye for this purpose is YO-PRO-1, which is a pro-fluorescent agent that intercalates with DNA. However, other dyes with varying molecular weights, such as ethidium, YOYO-1 or TOTO-1 [49,56], and carrying opposite charges (e.g., FITC) [27] have also been employed. 

The second methodology is whole-cell patch-clamp electrophysiology, which determines the relative permeabilities of different species passing through the P2X7-activated pore. In investigations of macropore formation, bi-ionic conditions have typically been employed, where *N*-methyl-d-glucamine (NMDG^+^), a large synthetic organic cation, and Na^+^ are the only permeating ions in the external and internal solutions, respectively. Upon ATP application, changes in the reversal potentials are observed over time and were initially interpreted as a progressive increase in the relative permeability of NMDG^+^ to Na^+^ [24,25]. This interpretation assumed that the concentrations of cations remained unchanged during the patch-clamp recordings. In 1999, the concept of pore dilation was proposed to explain the relative increase in permeability, suggesting that the channel undergoes progressive physical pore expansion [24,25]. However, in 2015, the group led by K. Swartz challenged this concept by demonstrating that significant changes in ion concentrations take place inside the cell during patch-clamp experiments, indicating that changes in reversal potentials cannot reliably measure a progressive increase in relative permeability [54]. Importantly, the study did not dispute the entry of large molecules into cells but necessitated a reevaluation of the mechanism of pore dilation. 

### 4.1. Pathway #1: The P2X7 Pore

As the concept of pore dilation has been discarded, the mechanism by which large molecules permeate the P2X7 pore has been a subject of recent investigation. Recent data provide evidence that some of these molecules, particularly the smallest ones, may directly permeate the P2X7 pore [26,46] (Figure 3B). The cryo-EM structure of the ATP-bound state of P2X7 has revealed an open pore diameter of approximately 5 Å, without any signs of channel dilation [40]. This aperture is wide enough to accommodate the permeation of small inorganic cations, such as partially hydrated Na^+^ ions, which have an ionic diameter of 1.9 Å, as expected for an open channel pore. 

However, for larger molecules such as NMDG^+^ or YO-PRO-1, the open diameter of the pore appears to reach minimum size (minimal cross-section of approximately 6 Å for NMDG^+^ and 7 Å for YO-PRO-1). This suggests that if these molecules pass through the pore, the process must occur at a much slower rate than that of Na^+^. Recent studies employing single-channel recordings, photoswitchable tweezers and modelling analyses provide support for this hypothesis [26,60,61,62]. These investigations have demonstrated that the entry of NMDG^+^ and other small cationic molecules, including spermidine, takes place immediately after ATP gating but at permeation rates approximately 10 times slower compared to Na^+^ ions [26,60,61,62]. 

In the case of YO-PRO-1, determining its intrinsic permeation rates through P2X7 is extremely challenging using patch-clamp electrophysiology. Dye uptake studies have shown that the apparent rates of YO-PRO-1 uptake are extremely slow, with the increase in fluorescence measured over minutes (Figure 3A). Consequently, currents carried by YO-PRO-1 are expected to be too small to be reliably measured by patch-clamp techniques. However, it is important to acknowledge that dye uptake is a complex process that involves various diffusional steps within the cell for the dye to reach the nucleus. These diffusional processes are likely the rate-limiting factor in fluorescence emission. 

Further support for the direct permeation of large molecules through P2X7 pores comes from experiments conducted with liposomes [46]. These studies involved the reconstitution of artificial DNA-encapsulated liposomes containing only the panda P2X7 receptor, without the presence of any other proteins. Upon ATP application to these liposomes, robust uptake of the YO-PRO-1 dye was observed, providing evidence that the pore is sufficiently large to allow the passage of YO-PRO-1 [46]. However, it is worth noting that these experiments utilized a truncated panda P2X7 construct lacking the N- and C-termini (referred to as P2X7-ΔNC), and the possibility of pore distortion cannot be ruled out. 

Molecules carrying a negative charge, such as fluorescein isothiocyanate (FITC), have also been proposed to directly pass through the P2X7 pore itself [27]. This finding is unexpected since P2X7 is known to be cation selective. However, it has been estimated that the entry of FITC occurs at significantly lower rates compared to chloride (Cl^−^) ions. Taking into account a relative permeability ratio (*P*_Cl_/*P*_Na_) of 0.01 for P2X7, the overall entry of anionic molecules would be extremely low [27]. 

Finally, structural analysis has revealed the presence of an electron density within the transmembrane helices, located at the near center of the plasma membrane [40]. This density is believed to correspond to a phospholipid moiety, with its headgroup positioned between two TM2 helices [40]. However, the specific role of this bound phospholipid in the permeation of small molecules, such as YO-PRO-1, and its influence on channel narrowing in the open state remain unclear. It is possible that this phospholipid bound to the P2X7 channel may limit the permeation of these molecules, and in its absence, a slight increase in the open diameter could favor the passage of larger molecules. 

Overall, these findings suggest that the pathway of YO-PRO-1 and comparable molecules is highly similar, if not identical, to that of small cations, including Ca^2+^, spermidine, and NMDG^+^.

### 4.2. Pathway #2: The P2X7-Scramblase Axis 

It has been suggested that YO-PRO-1, ethidium and larger molecules such as YOYO-1 or TOTO-1 can also enter cells through alternative pathways that are triggered by activated P2X7 [49,59,63]. Studies have proposed several potential pathways, including the involvement of Cl^−^ channels [49] and pannexin-1 [63]. However, evidence indicates that the contribution of pannexin-1 related to P2X7 macropore formation is not consistent across all cells [49,58], implying the existence of other unidentified additional pathways that may play a more universal role. Recent research has shed light on such new pathways involving a P2X7-scramblase axis [47,64] (Figure 3C). 

In 2015, the transmembrane protein TMEM16F was identified as a dual CaCC and phospholipid scramblase, playing a role in mediating essential effects downstream of P2X7 in mouse macrophages [64]. TMEM16F is a member of the TMEM16 (also known as Anoctamin), which consists of 10 proteins (TMEM16A-K, excluding I) [65]. While TMEM16A and TMEM16B function as CaCCs, the other paralogues are believed to act as phospholipid scramblases or dual-function non-selective ion channel/phospholipid scramblases, as seen in TMEM16F. X-ray and cryo-EM structures have revealed that these proteins form dimers, with each subunit composed of 10 transmembrane segments and coordinating two Ca^2+^ ions within the inner membrane leaflet [65]. Upon Ca^2+^ binding, TMEM16F’s scramblase activity disrupts phospholipids’ asymmetry. Consequently, phosphatidylserine (PS), a negatively charged phospholipid normally sequestered on the inner leaflet of the membrane bilayer under resting conditions, becomes exposed on the outer leaflet. The exposure of PS serves as a biological signal, triggering various processes including blood coagulation or cell death [66]. 

The study published in 2015 revealed the existence of a functional axis between P2X7 and TMEM16F in macrophages [64]. Activation of P2X7 leads to an influx of intracellular Ca^2+^, subsequently activating TMEM16F (Figure 3C). This activation leads to plasma membrane blebbing and apoptosis. The relationship between TMEM16F and macropore formation has been further supported by findings that pharmacological inhibition or genetic ablation of TMEM16F reduces YO-PRO-1 dye uptake in macrophages and HEK-293 cells [47,64]. Notably, the extent of inhibition varies between cell lines, with near-complete inhibition observed in macrophages [64] and only 40% inhibition in HEK-293 cells [47], indicating the existence of various routes for YO-PRO-1 entry into cells (putatively pathways #1 and #2 in Figure 3B,C). A close proximity of P2X7 and TMEM16F proteins, as suggested by co-immunoprecipitation experiments [47,64], would be instrumental in facilitating the efficient activation of the scramblase induced by P2X7 activation. However, it remains to be fully investigated whether P2X7 and TMEM16F physically interact. 

Another recent study has identified a scramblase called XK, which, together with the cytosolic lipid transporter VPS13A, acts as a prerequisite for P2X7-mediated phospholipid scrambling and cell lysis in mouse T cells [67]. The XK-VSP13A complex forms a protein complex at the plasma membrane, and upon P2X7 activation by ATP binding, it induces phospholipid scrambling, resulting in a novel form of necrosis [67,68]. 

These recent findings provide compelling evidence that P2X7 serves as a central receptor orchestrating the activation of multiple secondary pathways, including scramblases, which amplify the initial ATP binding signal (Figure 3C). These secondary pathways may represent the natural conduits for synthetic dyes that have been used to monitor cell permeation for over 40 years. However, the precise localization of these conduits is yet to be determined. It is thus plausible that larger molecules exceeding the diameter of the P2X7-open pore (≥5 Å) may transit through XK and TMEM16F scramblases via a mechanism that requires further investigation. We hypothesize that the C-terminus, including the ballast region, of P2X7 may play a role in the initiation of these secondary pathways. This is supported by experimental evidence demonstrating that the artificial deletion (P2X7∆C) or natural truncation (isoform P2X7B) of the C-terminus specifically impact dye uptake while minimally affecting the flux of small ions such as Ca^2+^ or Na^+^ [22,39,40]. Future experiments should focus on identifying the specific region of P2X7 that controls the activation of these secondary pathways.

## 5. A new Paradigm for P2X7-Mediated Switching of Cell Fate Decisions

P2X7 exhibits a switch from promoting cell proliferation to inducing cell death, although the exact mechanism behind this switch remains unknown [69]. Understanding this molecular switch is crucial as it could pave the way for the development of novel therapeutic interventions. It has been proposed that high levels of eATP, typically found at damaged sites, induce cell death, while low levels of eATP may favor cell survival [69]. 

The various models described in this review provide a plausible explanation for this molecular switch. At low eATP concentrations, the low activity of P2X7 allows for the permeation of small cations, such as Ca^2+^, in quantities that support cell survival (Figure 3B). This hypothesis is strongly supported by the observation that the isoform P2X7B, which lacks the C-terminus including the C-cys anchor, promotes cell proliferation by enabling the entry of Ca^2+^ [39]. However, it is worth noting that the removal of the C-cys anchor leads to channel desensitization [40], suggesting that the entry of Ca^2+^ may be limited in the P2X7B isoform. Conversely, at higher eATP concentrations or with prolonged eATP application, the increased activity of the pore triggers secondary pathways, including Cl^−^ channels and/or scramblases, which can activate cell death mechanisms (Figure 3C). It is speculated that both pathways may coexist within the same cell, as demonstrated in HEK-293 cells, where P2X7 activation induces YO-PRO-1 dye uptake through two distinct components: one involving TMEM16F (contributing approximately 40%) and the other likely occurring through the P2X7 pore itself (contributing approximately 60%) [47]. 

These findings suggest that the level of eATP and the resulting P2X7 activity play a critical role in determining cell fate, with low ATP levels favoring cell survival and high eATP levels triggering cell death through secondary pathways. Further research is needed to fully elucidate the interplay between these pathways and their contribution to the molecular switch observed in P2X7-mediated cell responses.

## 6. The Role of Macropore Formation and Current Facilitation in the P2X7-Associated Diseases

A relevant question is whether macropore formation and current facilitation contribute to P2X7-related diseases. While the precise extent of their contribution is not yet fully understood, existing evidence strongly supports that specific alterations in these features, particularly macropore formation, have functional implications. 

Over the past two decades, it has become evident that P2X7 plays a significant role in highly prevalent diseases. In the central nervous system, P2X7 has been implicated in neurodegenerative diseases [70,71], psychiatric disorders [72], neuropathic pain [73], epilepsy [71,74] and multiple sclerosis [71,75]. P2X7 has been shown to be upregulated in microglia surrounding the accumulation of amyloid beta (Aβ) peptides [76], a hallmark of Alzheimer’s disease (AD). In murine models, inhibiting or lacking P2X7 has been associated with a decrease in amyloid plaques and Aβ load [77,78,79]. Furthermore, the P2X7-dependent exacerbation of Huntington’s disease symptoms has been observed in murine models, and treatment with brilliant blue G, a P2X7 antagonist, leads to a reduction in the overall pattern of neurodegeneration. These findings suggest a generalized role of this receptor in neurodegenerative diseases [80]. However, the contribution of macropore formation and current facilitation to the progression of AD is not yet clear, and further studies are necessary. 

Regarding neuropathic pain, there is strong evidence supporting the contribution of macropore formation. Studies have shown that the naturally occurring P451L allelic mutation, located in the ballast (unfortunately not resolved in the single-particle cryo-EM structures), reduces sensitivity to ATP-induced macropore formation while not affecting channel function [81]. Haplotype mapping in mice has demonstrated that this mutation alleviates the effects of mechanical allodynia, suggesting that specifically targeting macropore formation while preserving cation channel activity could be a novel therapeutic strategy for reducing pain in individuals carrying *P2RX7* haplotypes that confer a high risk for chronic pain [12]. Numerous polymorphisms have now been described, some resulting in loss of function and others leading to gain of function [14]. 

In cancer, P2X7 has also been found to be involved in tumor progression [69,82]. Studies have provided insights into the relationship between the ion channel and the macropore features of P2X7 in vivo. The tumor microenvironment can contain sufficient levels of eATP for prolonged P2X7 activation [83], and it is suggested that these levels peak in localized microdomains near the plasma membrane [84]. Interestingly, it has been found that high levels of ATP in the tumor microenvironment might promote the expression of non-pore functional P2X7 (nfP2X7) or the truncated isoform P2X7B in tumor cells [85,86]. These P2X7 variants are fully uncoupled from macropore formation and lack cytotoxic activity but are still associated with a cytoplasmic increase in Ca^2+^, supporting proliferation. Moreover, the wild-type P2X7 is incapable of rescuing cells from death caused by knockdown of endogenous nfP2X7 [85]. This suggests the involvement of an intricate cancer cell survival mechanism that encompasses the ion channel function of P2X7 along with the inhibition of macropore activity. Additionally, these forms of P2X7 have been found to be overexpressed in tissue samples taken from patients compared to the control group [85]. 

In summary, the exact implications of the macropore formation in pathology are not yet fully understood; however, the available evidence strongly supports its significant involvement. To unravel the complexities of this intricate phenomenon, it is crucial to develop innovative approaches that enable the construction of a robust and comprehensive model incorporating molecular, cellular, and in vivo investigations. Disentangling the specific characteristics of individual P2X7 functions poses a tremendous challenge. Of particular significance is the Herculean task of deciphering P2X7-dependent macropore formation in pathological diseases, as isolating this feature from the ion channel activity proves exceptionally difficult, particularly in more complex models such as living animals.

## 7. Conclusions

In conclusion, this review provides an overview of our current understanding of P2X7 current facilitation and macropore formation. The recently determined full-length structures of P2X7 have provided valuable insights into the role of its cytoplasmic domain, particularly the significance of the C-cys anchor and unique ballast element in relation to current facilitation and macropore formation. Furthermore, the presence of a phospholipid, modeled as a PS [40], bound to the P2X7 and contributing to channel narrowing, raises intriguing possibilities, as its presence may be influenced by scramblases activated by P2X7, potentially serving as a regulatory mechanism. Further investigation is warranted to elucidate the precise functions of these elements in these mechanisms and to explore the impact of regulatory factors. Undoubtedly, these unanswered questions will continue to drive research in this field for years to come.

## Figures and Tables

**Figure 1 ijms-24-10896-f001:**
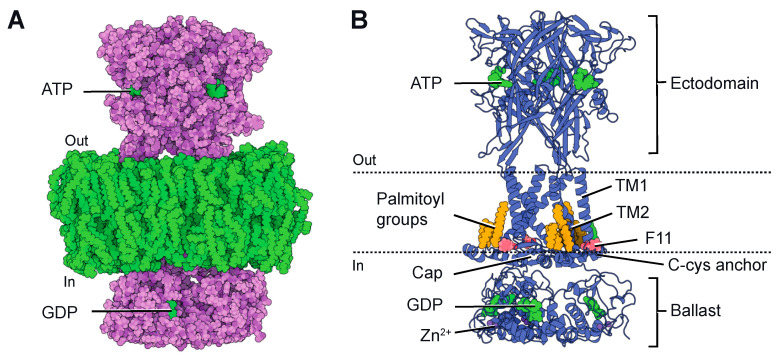
(**A**). Cryo-EM structure of rat P2X7 (pdb code: 6u9w) [40] embedded in a lipid bilayer, represented as spheres. The two nucleotide-binding sites, ATP and GDP, are indicated. (**B**). Ribbon representation of the same view, highlighting relevant structural features (ATP and GDP in green, Zn^2+^ in violet, palmitoyl groups in orange and F11 residue in red). Dotted lines depict the approximate position of the membrane boundaries. Structures were displayed using Protein Imager (https://3dproteinimaging.com/protein-imager/, accessed on 9 May 2023).

**Figure 2 ijms-24-10896-f002:**
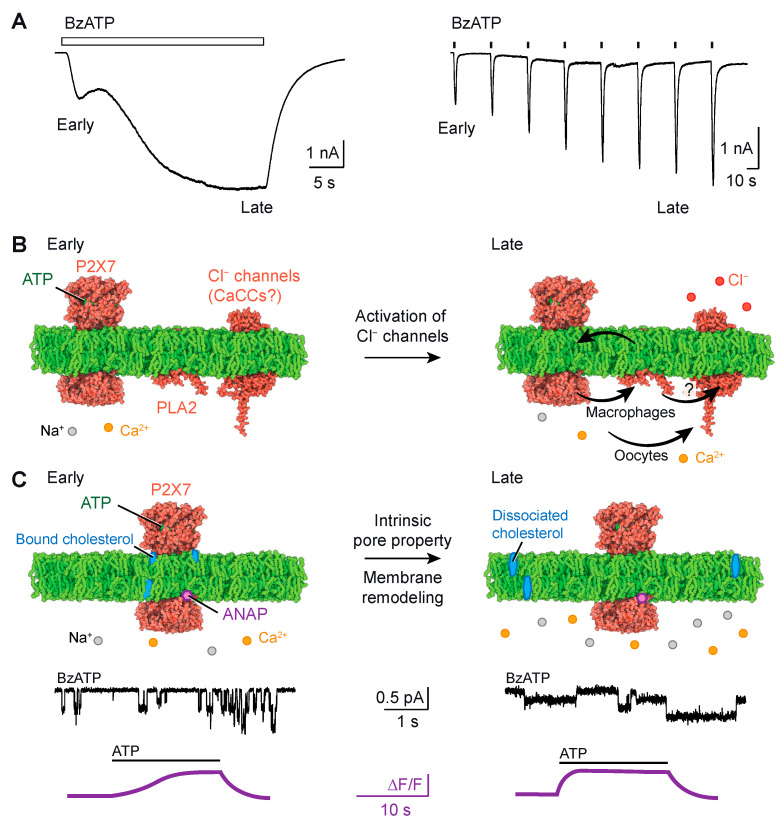
(**A**). Examples of current facilitation resulting from either the prolonged (30 s, left) or the repeated (2 s, right) application of 10 μM (2′(3′)-O-(4-benzoylbenzoyl)ATP) (BzATP), which is an ATP analogue that is more potent than ATP in activating P2X7. Notice the increase in inward current during the late stage compared to the early stage. Traces were adapted from [47]. (**B**,**C**). Proposed mechanistic models underlying current facilitation. (**B**). Model #1 suggests that current facilitation occurs through the secondary activation of Cl^−^ channels. This secondary activation can be mediated by Ca^2+^, functioning as a second messenger, as observed in oocytes [51] or through PLA2 (Q9NP80 predicted from AlphaFold [52]), in a Ca^2+^-independent manner, as demonstrated in macrophages [49]. The specific identity of the Cl^−^ channels remains unknown, but they could be CaCCs (here exemplified by Q4KMQ2 predicted from AlphaFold). (**C**). Model #2 suggests that current facilitation is an intrinsic property of P2X7 itself (pdb code: 6u9w [40]), as evidenced by an increase in open channel probability following a 30 s perfusion of 10 μM BzATP (recorded through single-channel currents [47]) or an acceleration of the ANAP fluorescent change upon a second 300 μM ATP application (recorded through voltage-clamp experiments [53], note the different time scales). It has been suggested that cholesterol exerts inhibitory effects on P2X7 activity by directly interacting with the transmembrane domain [45,46,47]. This implies that the dissociation of cholesterol from the receptor may alleviate its inhibition. Traces were adapted from [47,53]. Structures were displayed using Protein Imager (https://3dproteinimaging.com/protein-imager/, accessed on 9 May 2023).

**Figure 3 ijms-24-10896-f003:**
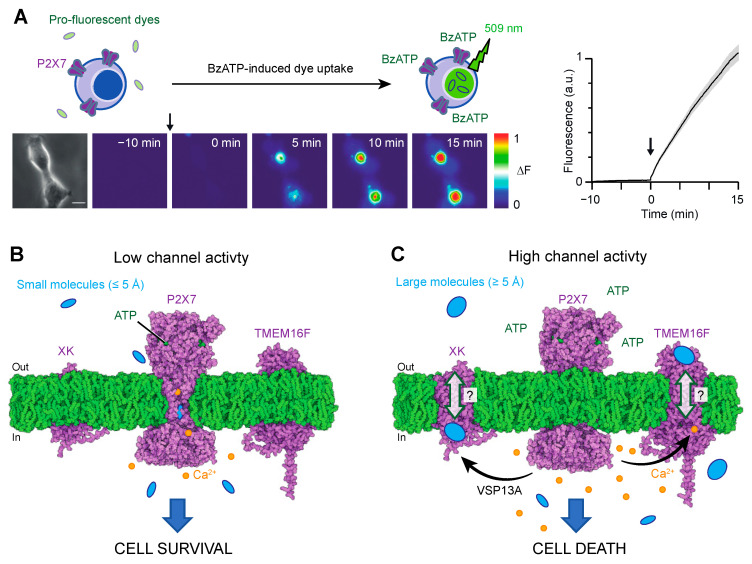
(**A**). Example of YO-PRO-1 dye uptake used to monitor macropore formation in HEK cells upon 10 μM BzATP application at time 0 (indicated by the arrow). Data were taken from [47]. (**B**,**C**). Proposed pathways involved in macropore formation. (**B**). Pathway #1: At low eATP concentrations, the low channel activity permits the passage of Na^+^ or Ca^2+^ ions, as well as small cations such as spermidine (145 Da), NMDG^+^ (195 Da), ethidium (314 Da) or YO-PRO-1 (376 Da) directly through the P2X7 pore itself, albeit at considerably slower rates. This slow channel activity, induced by low eATP concentration or during the initial stage of large ATP application (refer to Figure 2A), is suggested to favor cell survival. (**C**). Pathway #2: With higher eATP concentrations or prolonged activation, enhanced P2X7 pore activity triggers secondary scramblases, such as TMEM16F (Q4KMQ2 predicted from AlphaFold) or XK (P51811 predicted from AlphaFold), facilitating the passage of larger molecules, such as YOYO-1 (763 Da) or TOTO-1 (795 Da), for which their size may exceed the open diameter of P2X7 (≥5 Å). This mechanism responsible for the activation of the secondary pathway remains unknown, but it is speculated that the involvement of the ballast domain may play a role in this process. It should be noted that increased channel activity resulting from prolonged ATP application or high ATP concentrations further amplifies these pathways, which can contribute to cell death. Pathway #1 may coexist with pathway #2. Structures were displayed using Protein Imager (https://3dproteinimaging.com/protein-imager/, accessed on 9 May 2023).

## Data Availability

No new data were created.

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
