# Peer review of "Untangling Macropore Formation and Current Facilitation in P2X7"

_ijms, 2023, doi:10.3390/ijms241310896_

Round 1

Author Response

We thank the reviewers for their helpful comments. Below is a point-by-point reply.

Reviewer 1

This is a timely review of P2X7 receptor structure and function with a particular focus on sensitization of the receptor, permeation properties and downstream signalling mechanisms.

Response: Thank you.

There are places where the writing needs to be more precise and also where further details would be beneficial. I have summarized below.

  1. Make changes to the following sentence: “Unlike other P2X receptors, P2X7 does not undergo channel desensitization which a temporary inactivation that terminates ion flux despite the fact that ATP remains bound to the receptor”.

Response: We have made the change in the sentence as suggested. Now it reads as follows: “Unlike other P2X receptors, P2X7 typically does not undergo channel desensitization, which is a temporary inactivation that terminates ion flux despite ATP remaining bound to the receptor.”

  1. The meaning of the term heightened permeability within the following sentence is unclear. “While other P2X receptors also exhibit permeability to large organic cations [24-26], the distinct feature of P2X7 macropore formation lies in its heightened permeability, extending to nanometer-scale molecules”

Response: We have removed this term and rephrase the sentence. Now it reads as follows: “While other P2X receptors also exhibit permeability to organic cations [24-26], the distinct feature of P2X7 macropore formation lies in its ability to allow the passage of nanometer-scale molecules [27].”

  1. Model 1 “intervention of another channel” on page 5 need further explanation. The first part discusses a possible role for TMEM16A in current facilitation. How does a “significant and sustained increase in Cl- conductance in response to the Ca2+ influx induced by P2X7 opening” lead to an enhancement of inward current (example illustrated in Fig 1A), and a leftward shift in ATP concentration response relationship? Later in the paragraph there is a brief discussion of a role for PLA2 and release of phospholipids. What is the relationship between Cl channel activation and activation of PLA2? Fig 1B doesn’t make this clear.

Response: We now provide more details and have added sentences to explain how the increase in Cl- conductance leads to an enhancement of inward current. We have also added a few sentences related to the point raised by the reviewer.

Now it reads as follows: “Model #1 suggests that current facilitation occurs through the secondary activation of Cl- channels (Figure 2B). In heterologous systems such as Xenopus oocytes, the ectop-ic co-expression of P2X7 and TMEM16A (or anoactamin-1), a calcium-activated chloride channel (CaCC), leads to a significant and sustained increase in Cl- conductance in re-sponse to the Ca2+ influx induced by P2X7 opening [51]. It has been suggested that the increasing inward currents, recorded by two-electrode voltage-clamp electrophysiology, represent the combination of two major currents: the P2X7-mediated influx of Na+ and Ca2+, and the CaCC-mediated efflux of Cl- [51]. Both currents contribute to the down-ward deflection of the current trace, as the efflux of negative charges is electrically equivalent to the influx of positive charges. Similar results have been obtained in Axo-lotl oocytes, which lack endogenous CaCCs, thus ruling out the contribution of endoge-nous channels [51]. However, the impact of secondary activation of Cl- channels on the modulation in ATP sensitivity in P2X7 remains unknown. Furthermore, in other cell types like human macrophages, the role of Ca2+ as a second messenger in the second-ary activation of TMEM16A seems to be less significant. Instead, it has been demon-strated that current facilitation requires the involvement of Cl- channels through a phospholipase A2 (PLA2)-dependent pathway [49]. PLA2 is an enzyme that catalyzes the release of membrane phospholipids upon activation, and it operates independently of Ca2+. Certain phospholipids, including 1-palmitoyl-2-oleoyl-sn-glycero-3-phospho-(1'-rac-glycerol) (POPG) and sphingomyelin, have been shown to enhance P2X7 activity [46]. The released phospholipids from PLA2 activation can provide feedback to P2X7, resulting in current facilitation [49]. It is also possible that a similar mechanism affects the unidentified Cl- channel in macrophages. However, the relationship be-tween PLA2 and the Cl- channel is unclear and requires further investigation.”

  1. I suggest adding the words “inside the cell” to the following sentence to improve clarity: “However, in 2015, the group led by K.Swartz challenged this concept by demonstrating that significant changes in ion concentrations occur inside the cell during patch-clamp experiments, indicating that changes in Erev cannot reliably measure a progressive increase in relative permeability [51].”

Response: Thank you for the suggestion, we have added these words.

  1. The final paragraph on page 7 is unclear. I suggest simplifying and not directly comparing Yo-Pro with Na+ flux.

e.g. “Dye uptake studies suggest that the rates of YO-PRO-1 uptake are slow, with the increase in fluorescence measured on a time scale of minutes (Figure 3A). The size of the current carried by YO-PRO-1 is therefore expected to be too small to reliably measure by patch clamp. ”

At the end of the paragraph is written that “… the actual passage of YO- PRO-1 across the membrane itself, is believed to happen within seconds or less”. Again the meaning of this is unclear. How many YO-PRO-1 molecules are thought to move through the pore per second and how was this measured? If this is unknown then I would remove this.

Response: Thank you for the suggestions. We have changed the paragraph and modified the text as suggested. Now it reads as follows: “In the case of YO-PRO-1, determining its intrinsic permeation rates through P2X7 is extremely challenging using patch-clamp electrophysiology. Dye uptake studies have shown that the apparent rates of YO-PRO-1 uptake are extremely slow, with the increase in fluorescence measured over minutes (Figure 3A). Consequently, currents carried by YO-PRO-1 are expected to be too small to be reliably measured by patch-clamp techniques. However, it is important to acknowledge that dye uptake is a complex process that involves various diffusional steps within the cell for the dye to reach the nucleus. These diffusional processes are likely the rate-limiting factor in fluorescence emission.”

  1. The following paragraph comparing FITC uptake with Cl- is also confusing as written. iN the study referenced, Cl flux was measured only after the introduction of positive side chains by mutagenesis into the inner half of TM2.

Response: We appreciate this comment. However, if the reviewer looks carefully at the data (e.g. Figure 3 of Browne et al. J. Neurosci 2013), simultaneous dye uptake of positive and negative dyes was carried out in the wild-type P2X7 receptor. We agree that the authors also investigated FITC uptake in mutant receptors and showed that depending on the charge introduced by site-directed mutagenesis, this influences dye uptake, but they also provide evidence that this occurs in the wild-type receptor.

  1. Section 4.1, describes the P2X7-scramblase axis. The idea that “larger molecules exceeding the diameter of the P2X7-open pore (≥ 5 Å ) may transit through XK and TMEM16F scramblases” would benefit from a more detailed evaluation of the Ousingsawat et al and Dunning et al studies.

Response: As suggested, we have added more information. It now reads as follows: “The study published in 2015 revealed the existence of a functional axis between P2X7 and TMEM16F in macrophages [64]. Activation of P2X7 leads to an influx of intra-cellular Ca2+, subsequently activating TMEM16F (Figure 3C). This activation leads to plasma membrane blebbing and apoptosis. The relationship between TMEM16F and macropore formation has been further supported by findings that pharmacological in-hibition or genetic ablation of TMEM16F reduces YO-PRO-1 dye uptake in macrophag-es and HEK-293 cells [47, 64]. Notably, the extent of inhibition varies between cell lines, with near-complete inhibition observed in macrophages [64] and only 40% inhi-bition in HEK-293 cells [47], indicating the existence of various routes for YO-PRO-1 en-try into cells (putatively pathways #1 and #2 in Figure 3B and C). A close proximity of P2X7 and TMEM16F proteins, as suggested by co-immunoprecipitation experiments [47, 64], would be instrumental in facilitating efficient activation of the scramblase in-duced by P2X7 activation. However, it remains to be fully investigated whether P2X7 and TMEM16F physically interact.”

  1. The final section (6), on P2X7 associated diseases, does not add much to the review. I suggest expanding earlier sections and perhaps leaving this out.

Response: Although we recognize that there are not many examples where macropore formation is related to diseases, we would like to keep this section as it may direct future research direction.

Reviewer 2 Report

This manuscript  clearly and concisely presents accurate and completed data regarding the modern understanding of P2X7 macropore formation, current facilitation phenomemon and the receptor involvement in various pathologies. It might be published in IJMS journal in present form.

Minor suggestions:

P.1. remove extra point  "†These authors contribute equally. ."

P.2. To avoid repetition in the sentence "Although the first atomic resolution structure of a P2X structure was published in 2009 [29], the pioneering structures of P2X7 were not released until 2016",

authors might replace it to " Although the first atomic resolution structure of P2X receptor was published in 2009 [29], the pioneering structures of P2X7 were not released until 2016".

P.3. In the Figure 1. caption I suppose that " The two nucleotide-binding sites ATP and GDP, are indicated" instead of " The two nucleotide-binding sites, ATP and ADP, are indicated".

P.3. Replace the " juxtatransmembrane" to "juxta-transmembrane".

P.4. To improve the resolution of figure 2 if possible.

I think some titles looks like unfinished sentence or statement and need to be rephrased:

P.5. "3.2 Model #2: intrinsic property of P2X7"

P.9.  "5. Various models may provide a new paradigm for the P2X7 switch"

P.9. "6. Connections with P2X7-associated diseases"

For example it might be rephrased as

"5. A new paradigm for the P2X7-switching of cell fate decisions"

"6. The role of macropore formation and current facilitation in the P2X7-associated diseases"

Author Response

We thank the reviewers for their helpful comments. Below is a point-by-point reply.

Reviewer 2

This manuscript  clearly and concisely presents accurate and completed data regarding the modern understanding of P2X7 macropore formation, current facilitation phenomemon and the receptor involvement in various pathologies. It might be published in IJMS journal in present form.

Response: Thank you for your comment.

Minor suggestions:

P.1. remove extra point  "†These authors contribute equally. ."

Response: Done

P.2. To avoid repetition in the sentence "Although the first atomic resolution structure of a P2X structure was published in 2009 [29], the pioneering structures of P2X7 were not released until 2016", 

authors might replace it to " Although the first atomic resolution structure of P2X receptor was published in 2009 [29], the pioneering structures of P2X7 were not released until 2016".

Response: Thank you for pointing this out. We have modified the text as suggested.

P.3. In the Figure 1. caption I suppose that " The two nucleotide-binding sites ATP and GDP, are indicated" instead of " The two nucleotide-binding sites, ATP and ADP, are indicated".

Response: Thank you for pointing out this mistake. We have replaced ADP with GDP.

P.3. Replace the " juxtatransmembrane" to "juxta-transmembrane".

Response: Done

P.4. To improve the resolution of figure 2 if possible.

Response: We have uploaded all figures in higher resolution in the text.

I think some titles looks like unfinished sentence or statement and need to be rephrased:

P.5. "3.2 Model #2: intrinsic property of P2X7"

P.9.  "5. Various models may provide a new paradigm for the P2X7 switch"

P.9. "6. Connections with P2X7-associated diseases"

For example it might be rephrased as

"5. A new paradigm for the P2X7-switching of cell fate decisions"

"6. The role of macropore formation and current facilitation in the P2X7-associated diseases"

Response: Thank you for this comment. We have edited some titles as suggested.

“Title 3.1. Model #1: Current facilitation requires the involvement of an additional channel

Title 3.2. Model #2: Current facilitation is an intrinsic property of P2X7

Title 5. A new paradigm for P2X7-mediated switching of cell fate decisions

Title 6. The role of macropore formation and current facilitation in the P2X7-associated diseases”